# Learning to Play Sequential Games versus Unknown Opponents

**Pier Giuseppe Sessa**
ETH Zürich
sessap@ethz.ch

**Ilija Bogunovic**
ETH Zürich
ilijab@ethz.ch

**Maryam Kamgarpour**
ETH Zürich
maryamk@ethz.ch

**Andreas Krause**
ETH Zürich
krausea@ethz.ch

## Abstract

We consider a repeated sequential game between a learner, who plays first, and an opponent who responds to the chosen action. We seek to design strategies for the learner to successfully interact with the opponent. While most previous approaches consider known opponent models, we focus on the setting in which the opponent's model is *unknown*. To this end, we use *kernel-based* regularity assumptions to capture and exploit the structure in the opponent's response. We propose a novel algorithm for the learner when playing against an adversarial sequence of opponents. The algorithm combines ideas from bilevel optimization and online learning to effectively balance between *exploration* (learning about the opponent's model) and *exploitation* (selecting highly rewarding actions for the learner). Our results include algorithm's regret guarantees that depend on the regularity of the opponent's response and scale *sublinearly* with the number of game rounds. Moreover, we specialize our approach to repeated *Stackelberg games*, and empirically demonstrate its effectiveness in a traffic routing and wildlife conservation task.

## 1 Introduction

Several important real-world problems involve sequential interactions between two parties. These problems can often be modeled as two-player games, where the first player chooses a strategy and the second player responds to it. For example, in traffic networks, traffic operators plan routes for a subset of network vehicles (e.g., public transport), while the remaining vehicles (e.g., private cars) can choose their routes in response to that. The goal of the first player in these games is to find the optimal strategy (e.g., traffic operators seek the routing strategy that minimizes the overall network's congestion, *cf.*, [20]). Several algorithms have been previously proposed, successfully deployed, and used in domains such as urban roads [17], airport security [29], wildlife protection [40], and markets [15], to name a few.

In many applications, complete knowledge of the game is not available and, thus, finding a good strategy for the first player becomes more challenging. The response function of the second player, that is, how the second player responds to strategies of the first player, is typically unknown and can only be inferred by repeatedly playing and observing the responses and game outcomes [22, 6]. Consequently, we refer to the first and second players as *learner* and *opponent*, respectively. An additional challenge for the learner in such repeated games lies in facing a potentially different *type* of opponent at every game round. In various domains (e.g., in security applications), the learner can even face an adversarially chosen sequence of opponent/attacker types [4].

Motivated by these important considerations, we study a repeated sequential game against an *unknown* opponent with multiple types. We propose a novel algorithm for the learner when facing an adversarially chosen sequence of types. *No-regret* guarantees of our algorithm in these settings ensure that the learner's performance converges to the optimal one in hindsight (i.e., the idealized scenario in which the types' sequence and opponent's response function are known ahead of time).

To that end, our algorithm learns the opponent's response function online, and gradually improves the learner's strategy throughout the game.

**Related work.** Most previous works consider sequential games where the goal is to play against a *single* type of opponent. Authors of [22] and [28] show that an optimal strategy for the learner can be obtained by observing a polynomial number of opponent's responses. In security applications, methods by [34] and [19] learn the opponent's response function by using PAC-based and decision-tree behavioral models, respectively. Recently, single opponent modeling has also been studied in the context of deep reinforcement learning, e.g., [14, 30, 36, 13]. While all these approaches exhibit good empirical performance, they do not consider multiple types of opponents and lack regret guarantees.

Playing against *multiple* types of opponents has been considered in Bayesian Stackelberg games [27, 16, 25], where the opponent's types are drawn from a known probability distribution. Similarly, several type-based reasoning algorithms have been proposed (see, e.g., [1, Section 4.2]) which maintain beliefs over a set of possible opponent types, but they typically apply to non-sequential games and have few theoretical justifications. In [5], the authors propose no-regret algorithms when a set of opponents' behavioral models is available to the learner. In this work, we make no such distributional or availability assumptions, and our results hold for *adversarially* selected sequences of opponent's types. This is similar to the works [4, 39], in which the authors propose no-regret online learning algorithms to play repeated Stackelberg games [38]. In contrast to [4], we consider a more challenging setting in which opponents' utilities are *unknown* and focus on learning the opponent's response function from observing the opponent's responses. This is also different from the approach of [39], which is specifically designed to play security games (e.g., assign resources to protect targets) and does not use the observed data to learn the opponent's response function.

**Contributions.** Our main contributions are as follows:

- We propose STACKELUCB, a novel algorithm for playing sequential games versus an *adversarially* chosen sequence of opponent's types. Moreover, we also specialize our approach to the case in which the *same* type of opponent is faced at every round.
- We model the correlation present in the opponent's responses via kernel-based regularity assumptions, and prove the first *sublinear* kernel-based regret bounds.
- We consider repeated *Stackelberg games* with *unknown* opponents, and specialize our approach and regret bounds to this class of games.
- Finally, we experimentally validate the performance of our algorithms in *traffic routing* and *wildlife conservation* tasks, where they consistently outperform other baselines.

## 2 Problem Setup

We consider a sequential two-player repeated game between the learner and its opponent. The set of actions that are available to the learner and opponent in every round of the game are denoted by $\mathcal{X}$ and $\mathcal{Y}$, respectively. The learner seeks to maximize its reward function $r(x, y)$ that depends on actions played by both players, $x \in \mathcal{X}$ and $y \in \mathcal{Y}$. In every round of the game, the learner can face an opponent of different type $\theta_t \in \Theta$ that is unknown to the learner at the decision time. As the sequence of opponent's types can be chosen adversarially, we focus on randomized strategies for the learner as explained below. We summarize the protocol of the repeated sequential game as follows.

In every game round $t$:

1. The learner computes a randomized strategy $\mathbf{p}_t$, i.e., a probability distribution over $\mathcal{X}$, and samples action $x_t \sim \mathbf{p}_t$.
2. The opponent observes $x_t$ and responds by selecting $y_t = b(x_t, \theta_t)$, where $b : \mathcal{X} \times \Theta \to \mathcal{Y}$ represents the opponent's *response function*.
3. The learner observes the opponent's type $\theta_t$ and response $y_t$, and receives reward $r(x_t, y_t)$.

The opponent's types $\{\theta_i\}_{i=1}^T$ can be chosen by an *adaptive* adversary, i.e., at round $t$, the type $\theta_t$ can depend on the sequence of randomized strategies $\{\mathbf{p}_i\}_{i=1}^t$ of the learner and on the previous realized actions $x_1, \ldots, x_{t-1}$ (but not on the current action $x_t$). The goal of the learner is to maximize the cumulative reward $\sum_{t=1}^T r(x_t, y_t)$ over $T$ rounds of the game. We assume that the learner knows its reward function $r(\cdot, \cdot)$, while the opponent's response function $b(\cdot, \cdot)$ is unknown. To achieve this goal, the learner has to repeatedly play the game and learn about the opponent's response function

from the received feedback. After $T$ game rounds, the performance of the learner is measured via the cumulative regret:

$$R(T) = \max_{x \in \mathcal{X}} \sum_{t=1}^{T} r(x, b(x, \theta_t)) - \sum_{t=1}^{T} r(x_t, y_t). \tag{1}$$

The regret represents the difference between the cumulative reward of a single best action from $\mathcal{X}$ and the sum of the obtained rewards. An algorithm is said to be *no-regret* if $R(T)/T \to 0$ as $T \to \infty$.

**Regularity assumptions.** Attaining sub-linear regret is not possible in general for arbitrary response functions and domains, and hence, this requires further regularity assumptions. We consider a finite set of actions $\mathcal{X} \subset \mathbb{R}^d$ available to the learner, and a finite set of opponent's types $\Theta \subset \mathbb{R}^p$. We assume the unknown response function $b(x, \theta)$ is a member of a reproducing kernel Hilbert space $\mathcal{H}_k$ (RKHS), induced by some *known* positive-definite kernel function $k(x, \theta, x', \theta')$. RKHS $\mathcal{H}_k$ is a Hilbert space of (typically non-linear) well-behaved functions $b(\cdot, \cdot)$ with inner product $\langle \cdot, \cdot \rangle_k$ and norm $\| \cdot \|_k = \langle \cdot, \cdot \rangle_k^{1/2}$, such that $b(x, \theta) = \langle b(\cdot, \cdot), k((\cdot, \cdot), (x, \theta)) \rangle_k$ for every $x \in \mathcal{X}, \theta \in \Theta$ and $b \in \mathcal{H}_k$. The RKHS norm measures smoothness of $b$ with respect to the kernel function $k$ (it holds $\|b\|_k < \infty$ iff $b \in \mathcal{H}_k$). We assume a known bound $B > 0$ on the RKHS norm of the unknown response function, i.e., $\|b\|_k \leq B$. This assumption encodes the fact that similar opponent types and strategies of the learner lead to similar responses. This similarity is measured by the known kernel function that satisfies $k(x, \theta, x', \theta') \leq 1$ for any feasible inputs.[1] Most popularly used kernel functions that we also consider are linear, squared-exponential (RBF) and Matérn kernels [31].

Our second regularity assumption is regarding the learner's reward function $r : \mathcal{X} \times \mathcal{Y} \to [0, 1]$, which we assume is $L_r$-Lipschitz continuous with respect to $\| \cdot \|_1$.

## 3 Proposed Approach

The observed opponent's response can often contain some observational noise, e.g., in wildlife protection (see Section 4.2), we only get to observe an imprecise/inexact poaching location. Hence, instead of directly observing $b(x_t, \theta_t)$ at every round $t$, the learner receives a noisy response $y_t = b(x_t, \theta_t) + \epsilon_t$. For the sake of clarity, we consider the case of scalar responses, i.e., $y_t \in \mathbb{R}$, but in Appendix A, we also consider the case of vector-valued responses. We let $\mathcal{H}_t = \{\{(x_i, \theta_i, y_i)\}_{i=1}^{t-1}, (x_t, \theta_t)\}$, and assume $\mathbb{E}[\epsilon_t | \mathcal{H}_t] = 0$ and $\epsilon_t$ is conditionally $\sigma$-sub-Gaussian, i.e., $\mathbb{E}[\exp(\zeta \epsilon_t) | \mathcal{H}_t] \leq \exp(\zeta^2 \sigma^2 / 2)$ for any $\zeta \in \mathbb{R}$.

At every round $t$, by using the previously collected data $\{(x_i, \theta_i, y_i)\}_{i=1}^{t-1}$, we can compute a mean estimate of the opponent's response function via standard kernel ridge regression. This can be obtained in closed-form as:

$$\mu_t(x, \theta) = k_t(x, \theta)^T (K_t + \lambda I_t)^{-1} \boldsymbol{y}_t, \tag{2}$$

where $\boldsymbol{y}_t = [y_1, \ldots, y_t]^T$ is the vector of observations, $\lambda > 0$ is a regularization parameter, $k_t(x, \theta) = [k(x, \theta, x_1, \theta_1), \ldots, k(x, \theta, x_t, \theta_t)]^T$ and $[K_t]_{i,j} = k(x_i, \theta_i, x_j, \theta_j)$ is the kernel matrix. We also note that $\mu_t(\cdot, \cdot)$ can be seen as the posterior mean function of the corresponding Bayesian Gaussian process model [31]. The variance of the proposed estimator can be obtained as:

$$\sigma_t^2(x, \theta) = k(x, \theta, x, \theta) - k_t(x, \theta)^T (K_t + \lambda I_t)^{-1} k_t(x, \theta). \tag{3}$$

Moreover, we can use (2) and (3) to construct upper and lower confidence bound functions:

$$\text{ucb}_t(x, \theta) := \mu_t(x, \theta) + \beta_t \sigma_t(x, \theta), \quad \text{lcb}_t(x, \theta) := \mu_t(x, \theta) - \beta_t \sigma_t(x, \theta), \tag{4}$$

respectively, for every $x \in \mathcal{X}, \theta \in \Theta$, where $\beta_t$ is a confidence parameter. A standard result from [2, 35] (see Lemma 4 in Appendix A) shows that under our regularity assumptions, $\beta_t$ can be set such that, with high probability, response $b(x, \theta) \in [\text{lcb}_t(x, \theta), \text{ucb}_t(x, \theta)]$ for every $(x, \theta) \in \mathcal{X} \times \Theta$ and $t \geq 1$.

Finally, before moving to our main results, we define a sample complexity parameter that quantifies the *maximum information gain* about the unknown function from noisy observations:

$$\gamma_t := \max_{\{(x_i, \theta_i)\}_{i=1}^t} 0.5 \log \det(I_t + K_t / \lambda). \tag{5}$$

**Algorithm 1** The STACKELUCB algorithm (Playing vs. Sequence of Unknown Opponents)

---

**Input:** Finite action set $\mathcal{X} \subset \mathbb{R}^d$, kernel $k(\cdot, \cdot)$, parameters $\lambda, \{\beta_t\}_{t \geq 1}, \eta$

1: Initialize: Uniform strategy $\mathbf{p}_1 = \frac{1}{|\mathcal{X}|}\mathbf{1}_{|\mathcal{X}|}$
2: **for** $t = 1, 2, \ldots, T$ **do**
3:     Sample action $x_t \sim \mathbf{p}_t$              `// Opponent` $\theta_t$ `observes` $x_t$ `and computes` $b(x_t, \theta_t)$
4:     Observe $\theta_t$ and noisy response $y_t = b(x_t, \theta_t) + \epsilon_t$
5:     Compute optimistic reward estimates:
            $\forall x \in \mathcal{X} : \tilde{r}_t(x, \theta_t) := \max_y r(x, y), \quad$ s.t. $\ y \in \big[\text{lcb}_t(x, \theta_t), \text{ucb}_t(x, \theta_t)\big]$
6:     Perform strategy update: $\forall x \in \mathcal{X} : \ \mathbf{p}_{t+1}[x] \propto \mathbf{p}_t[x] \cdot \exp\big(\eta \cdot \tilde{r}_t(x, \theta_t)\big)$
7:     Update: $\mu_{t+1}, \sigma_{t+1}$ with $\{(x_t, \theta_t, y_t)\}$ (via (2), (3)), and $\text{ucb}_{t+1}, \text{lcb}_{t+1}$ (via (4))
8: **end for**

---

It has been introduced by [35] and later on used in various theoretical works on Bayesian optimization. Analytical bounds that are sublinear in $t$ are known for popularly used kernels [35], e.g., when $\mathcal{X} \times \Theta \subset \mathbb{R}^d$, we have $\gamma_t \leq \mathcal{O}(\log(t)^{d+1})$ and $\gamma_t \leq \mathcal{O}(d\log(t))$ for squared exponential and linear kernels, respectively. This quantity characterizes the regret bounds obtained in the next sections.

### 3.1 The STACKELUCB Algorithm

The considered problem (Section 2) can be seen as an instance of adversarial online learning [8] in which an adversary chooses a reward function $r_t(\cdot)$ in every round $t$, while the learner (without knowing the reward function) selects action $x_t$ and subsequently receives reward $r_t(x_t)$. To achieve *no-regret*, the learner needs to maintain a probability distribution over the set $\mathcal{X}$ of available actions and play randomly according to it. Recall that we consider a finite set of actions $\mathcal{X} \subset \mathbb{R}^d$ and we let $\mathbf{p}_t$ denote the probability distribution (vector) supported on $\mathcal{X}$. At every round, the learner then plays action $x_t \sim \mathbf{p}_t$ and subsequently updates its strategy to $\mathbf{p}_{t+1}$.

*Multiplicative Weights* (MW) [24] algorithms such as EXP3 [3] and HEDGE [12] are popular no-regret methods for updating $\mathbf{p}_t$, depending on the feedback available to the learner in every round. The former only needs observing reward of the played action $r_t(x_t)$ (*bandit* feedback), while the latter requires access to the entire reward function $r_t(\cdot)$ at every $t$ (*full-information* feedback).

The considered game setup corresponds (from the learner's perspective) to the particular online learning problem in which $r_t(\cdot) := r(\cdot, b(\cdot, \theta_t))$, type $\theta_t$ is revealed, and the bandit observation $y_t$ is observed by the learner. Full-information feedback, however, is not available as $b(\cdot, \theta_t)$ is unknown. To alleviate this, similarly to [32], we compute "optimistic" reward estimates to emulate the full-information feedback. Based on previously observed data, we establish upper and lower confidence bounds $\text{ucb}_t(\cdot)$ and $\text{lcb}_t(\cdot)$, of the opponent's response function (via (4) and Lemma 4, Appendix A). These are then used to estimate the optimistic rewards of the learner for any $x \in \mathcal{X}$ at round $t$ as:

$$\tilde{r}_t(x, \theta_t) := \max_y \ r(x, y) \quad \text{s.t.} \quad y \in \big[\text{lcb}_t(x, \theta_t), \text{ucb}_t(x, \theta_t)\big]. \tag{6}$$

We note that the learner's reward function is assumed to be known and that can be efficiently optimized for any fixed $x$. The latter assumption is realistic given that in many applications the learner can often choose its own objective (see examples in Section 4). For example, in case $r(x, \cdot)$ is a concave function, the problem in (6) corresponds to concave function maximization subject to convex constraints which can be performed efficiently via standard gradient-based methods. Optimistic rewards allow the learner to control the maximum incurred regret, while Lipschitness of $r(\cdot)$ ensures that learning the opponent's response function (via (2) and (3)) translates to more accurate reward estimates.

We are now in position to desribe our novel STACKELUCB algorithm for the learner (see Algorithm 1). STACKELUCB maintains a distribution $\mathbf{p}_t$ over $\mathcal{X}$, and samples actions $x_t \sim \mathbf{p}_t$ at every round. It maintains confidence bounds of the opponent's response function $b(\cdot, \cdot)$ by using the previously obtained opponent's responses (via (2)-(3)). For each $x \in \mathcal{X}$, optimistic rewards $\tilde{r}(x, \theta_t)$ are computed via (6) and used to emulate the full-information feedback. Finally, the distribution $\mathbf{p}_t$ is updated by the standard MW update rule: $\mathbf{p}_{t+1}[x] \propto \mathbf{p}_t[x] \cdot \exp\big(\eta \cdot \tilde{r}(x, \theta_t)\big)$, where $\eta$ is the learning step set as in the following theorem.

**Theorem 1** *Consider the setting with multiple opponent types from $\Theta$, and assume the learner's reward function is $L_r$-Lipschitz continuous. Then for any $\delta \in (0, 1)$, the regret of STACKELUCB*

*when used with $\lambda \geq 1$, $\beta_t = \sigma\lambda^{-1}\sqrt{2\log\left(\frac{1}{\delta}\right) + \log(\det(I_t + K_t/\lambda))} + \lambda^{-1/2}B$, and learning step $\eta = \sqrt{8\log(|\mathcal{X}|)/T}$, is bounded, with probability at least $1 - 2\delta$, by*

$$R(T) \leq \sqrt{\tfrac{1}{2}T\log|\mathcal{X}|} + \sqrt{\tfrac{1}{2}T\log\left(\tfrac{1}{\delta}\right)} + 4L_r\beta_T\sqrt{T\lambda\gamma_T},$$

*where $B \geq \|b\|_{\mathcal{H}_k}$ and $\gamma_T$ is the maximum information gain defined in (5).*

The obtained regret bound scales sublinearly with $T$, and depends on the regret obtained from playing HEDGE (first two terms) and learning of the opponent's response function (last term in the regret bound). We note that EXP3 attains $\mathcal{O}(\sqrt{T|\mathcal{X}|\log|\mathcal{X}|})$ while HEDGE attains improved $\mathcal{O}(\sqrt{T\log|\mathcal{X}|})$ regret bound which scales favourably with the number of available actions $|\mathcal{X}|$. The same holds for our algorithm, but crucially – unlike HEDGE – our algorithm uses the bandit feedback only.

Next, we consider a special case of a single opponent type, while in Section 3.3, we show how STACKELUCB can be used to play unknown repeated Stackelberg games.

## 3.2 Single Opponent Type

We now consider the special case where the learner is playing against the opponent of a single *known* type at every round of the game, i.e., $\theta_t = \bar{\theta}$. The goal of the learner is to compete with the action that is the solution of the following problem:

$$\max_{x\in\mathcal{X}} r(x, y) \quad \text{s.t.} \quad y = b(x, \bar{\theta}). \tag{7}$$

Even in this simpler setting, the learner cannot directly optimize (7), since the opponent's response function $b(\cdot, \bar{\theta})$ is unknown, and can only be inferred by repeatedly playing the game and observing its outcomes. The problem in (7) is a special instance of *bilevel optimization* [33] in which the lower-level function is unknown.

Next, we show that the learner can achieve no-regret by using the estimator, used in STACKELUCB, from (6), and following a simple yet effective strategy. At every round $t$, it consists of using the past observed data $\{(x_i, y_i, \bar{\theta})\}_{i=1}^{t-1}$ to build the confidence bounds as in (4), and selecting the action that maximizes the optimistic reward:

$$x_t = \arg\max_{x\in\mathcal{X}} \tilde{r}_t(x, \bar{\theta}). \tag{8}$$

This bilevel strategy is reminiscent of the *single level* GP-UCB algorithm used in standard Bayesian optimization [35], and leads to the following guarantee:

**Corollary 2** *Consider the setting where the learner plays against the same opponent $\bar{\theta} \in \Theta$ in every game round, and assume the learner's reward function is $L_r$-Lipschitz continuous. Then for any $\delta \in (0,1)$, the regret of the learner when playing according to (8) with $\beta_t$ set as in Theorem 1 and $\lambda \geq 1$, is bounded with probability at least $1 - \delta$ by*

$$R(T) \leq 4L_r\beta_T\sqrt{T\lambda\gamma_T},$$

*where $\|b\|_{\mathcal{H}_k} \leq B$ and $\gamma_T$ is the maximum information gain as defined in (5).*

The obtained bilevel regret rate is a constant factor $L_r$ worse in comparison to the rate of the standard single-level bandit optimization [35], and reflects the additional dependence of the learner's reward function on the opponent's response. Moreover, it shows that in the case of a single opponent the learner can achieve better regret guarantees compared to Theorem 1. Finally, we note that one could also consider modeling and optimizing $g(\cdot) = r(\cdot, b(\cdot, \bar{\theta}))$ directly (as a single unknown objective), but this can lead to worse performance as reasoned and empirically demonstrated in Section 4.2.

## 3.3 Learning in Repeated Stackelberg Games

We consider Stackelberg games [38] and show how they can be mapped to our general problem setup from Section 2. A Stackelberg game is played between two players: the *leader*, who plays first, and the *follower* who *best-responds* to the leader's move.[2] Moreover, in a *repeated* Stackelberg game (e.g., [22, 25]), leader and follower play repeated rounds, while the leader can (as before) face a

potentially different type of follower at every round [4]. In Stackelberg games, at every round the leader commits to a *mixed strategy* (i.e., a probability distribution over the actions): If we let $n_l$ be the number of actions available to the leader, we can map repeated Stackelberg games to our setup by letting $x_t \in \mathcal{X} = \Delta^{n_l}$ be the leader's mixed strategy at time $t$, where $\Delta^{n_l}$ stands for $n_l$-dimensional simplex. [3] Moreover, the opponent's response function in a Stackelberg game assumes the specific *best-response* form $b(x_t, \theta_t) = \arg\max_{y \in \mathcal{Y}} U_{\theta_t}(x_t, y)$, where $U_{\theta_t}(x, y)$ represents the expected utility of the follower of type $\theta_t$ under the leader's mixed strategy $x$ (as in [4] we assume the follower breaks ties in an arbitrary but consistent manner so that $b(x, \theta_t)$ is a singleton). We note that our regularity assumptions of Section 2 enforce smoothness in the follower's best-response and indirectly depend on the structure of the function $U_\theta(\cdot, \cdot)$ and on the follower's decision set $\mathcal{Y}$ (similar regularity conditions are used in other works on bilevel optimization, e.g., [11, 23]). Without further assumptions on the follower's types (see, e.g., [27, 16] for Bayesian type assumptions), the goal of the leader is to obtain sublinear regret as defined in Eq. (1).

Our approach is inspired by [4], where the authors consider the case in which the leader has complete knowledge of the set of possible follower types $\Theta$ and utilities $U_\theta(\cdot, \cdot)$ and show that it can achieve no-regret by considering a carefully constructed (via discretization) finite subset of mixed strategies. In this work, we consider the more challenging scenario in which these utilities are *unknown* to the leader and hence the follower's response function can only be learned throughout the game. Moreover, differently from [4], we consider infinite action sets available to the follower. Under our regularity assumptions, we show that the leader can attain no-regret by using STACKELUCB over a discretized mixed strategy set.

We let $\mathcal{D}$ be the finite discretization (uniform grid) of the leader's mixed strategy space $\Delta^{n_l}$ with size $|\mathcal{D}| = (L_r(1 + L_b)\sqrt{n_l T})^{n_l}$ chosen such that:

$$\|x - [x]_{\mathcal{D}}\|_1 \leq (L_r(1 + L_b))^{-1}\sqrt{n_l / T}, \quad \forall x \in \Delta^{n_l}, \tag{9}$$

where $[x]_{\mathcal{D}}$ is the closest point to $x$ in $\mathcal{D}$. Before stating the main result of this section, we further assume that the follower's response function $b(\cdot, \cdot)$ is $L_b$-Lipschitz continuous, so that differences in the follower's responses can be bounded in $\mathcal{D}$.[4]

**Corollary 3** *Consider a repeated Stackelberg game with $n_l$ actions available to the leader. Let the leader use STACKELUCB with $\mathcal{D}$ from (9) to sample a mixed strategy at every round. Then for any $\delta \in (0, 1)$, when STACKELUCB is run with $\lambda \geq 1$, $\beta_t$ is set as in Theorem 1 and $\eta = \sqrt{8\log(|\mathcal{D}|)/T}$, the regret of the leader is bounded, with probability at least $1 - 2\delta$, by*

$$R(T) \leq \sqrt{\frac{1}{2}Tn_l \log\left(L_r(1 + L_b)\sqrt{n_l T}\right)} + \sqrt{Tn_l} + \sqrt{\frac{1}{2}T \log\left(\frac{1}{\delta}\right)} + 4L_r\beta_T\sqrt{T\lambda\gamma_T}.$$

Compared to the $\mathcal{O}\left(\sqrt{T \cdot \text{poly}(n_l, n_f, k_f)}\right)$ regret of [4] ($n_f$ and $k_f$ are the numbers of actions available to the follower and possible follower types, respectively), our regret bound also scales sublinearly with $T$ and, unlike the result of [4], it holds when playing against followers with unknown utilities (also, potentially infinite number of follower types). The last term in our regret bound can be interpreted as the price of not knowing such utilities ahead of time. We remark that while both ours and [4]'s approaches are no-regret, they are both computationally inefficient since the number of considered mixed strategies (e.g., in Line 6 of Algorithm 1) is exponential in $n_l$.

## 4 Experiments

In this section, we evaluate the proposed algorithms in traffic routing and wildlife conservation tasks.

### 4.1 Routing Vehicles in Congested Traffic Networks

We use the road traffic network of Sioux-Falls [21], which can be represented as a directed graph with 24 nodes and 76 edges $e \in E$. We consider the traffic routing task in which the goal of the network operator (e.g., the local traffic authority) is to route 300 units (e.g., a fleet of autonomous vehicles) between the two nodes of the network (depicted as blue and green nodes in Figure 1). At the same time, the goal of the operator is to avoid the network becoming overly congested. We model

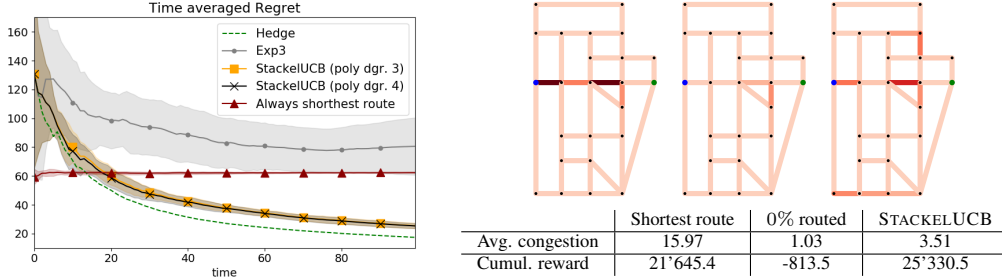

| | Shortest route | 0% routed | STACKELUCB |
|---|---|---|---|
| Avg. congestion | 15.97 | 1.03 | 3.51 |
| Cumul. reward | 21'645.4 | -813.5 | 25'330.5 |

Figure 1: **Left:** Time-averaged regret of the operator using different routing strategies. STACKELUCB (polynomial kernels of degree 3 or 4) leads to a smaller regret compared to the considered baselines and performs comparably to the idealized HEDGE algorithm. **Right:** Edges' congestion (color intensity proportional to the time-averaged congestion computed as in Appendix E) when the operator at each round: (left) Routes 100% of the units via the shortest route, (middle) Routes 0% of units, and (right) Uses STACKELUCB. When 100% of the units are routed via the shortest route the central edges are extremely congested. The congestion is reduced with STACKELUCB because alternative routes are selected. We report the respective average congestion levels and operator's cumulative rewards in the table.

this problem as a repeated sequential game (as defined in Section 2) between the network operator (learner) and the rest of the users present in the network (opponent). We evaluate the performance of the operator when using STACKELUCB to select routes.

We consider a finite set $\mathcal{X}$ of possible routing plans for the operator (generated as in Appendix E). At each round $t$, the routing plan chosen by the network operator can be represented by the vector $x_t \in \mathbb{R}_{\geq 0}^{|E|}$, where $x_t[i]$ represents units that are routed through edge $i \in E$. We let the *type* vector $\theta_t \in \mathbb{R}_{\geq 0}^{552}$ represent the demand profile of the network users at round $t$, where each entry indicates the number of users that want to travel between any pair (552 pairs in total) of nodes in the network. The network users observe the operator's routing plan $x_t$ and choose their routes according to their preferences. This results in a certain congestion level of the network. We represent such level as the average congestion of the edges $y_t = b(x_t, \theta_t) \in \mathbb{R}_+$, where $b(\cdot, \cdot)$ captures both the users' preferences and the network's congestion model (see Appendix E for details) and is *unknown* to the operator.

Given routing plan $x_t$ and congestion $y_t$, we use the following reward function for the operator: $r(x_t, y_t) = g(x_t) - \kappa \cdot y_t$, where $g(x_t)$ represents the total number of units routed to the operator's destination node at round $t$ and $\kappa > 0$ stands for a trade-off parameter. This parameter balances the two opposing objectives of the operator, i.e., routing a large number of units versus decreasing the overall network congestion. At the end of each round, the operator observes $y_t$ and $\theta_t$ and updates the routing strategy. Network's data and congestion model are based on [21], and a detailed description of our experimental setup is provided in Appendix E.

We compare the performance of the network operator when using STACKELUCB with the ones achieved by 1) routing 100% of the units via the shortest route at every round, 2) routing 0% of the units at every round, 3) the EXP3 algorithm and 4) the HEDGE algorithm. In this case, HEDGE corresponds to the algorithm by [4] and represents an *unrealistic* benchmark because the full-information feedback is not available to the network operator since the function $b(\cdot, \cdot)$ is unknown. We run STACKELUCB with polynomial kernels of degree 3 or 4 (polynomial functions are typically used as good congestion models, *cf.*, [21]), set $\eta$ according to Theorem 1 and use $\beta_t = 0.5$ (we also observed, as in [35], that theory-informed values for $\beta_t$ are overly conservative). Kernel hyperparameters are computed offline via maximum-likelihood over 100 randomly generated points.

STACKELUCB leads to a significantly smaller regret compared to the considered baselines, as shown in Figure 1 (the regret of baseline 2 is above the y-axis limit), and its performance is comparable to the full-information HEDGE algorithm. Moreover, we report the cumulative reward obtained by the operator when using STACKELUCB and other two baselines, together with the resulting time-averaged congestion levels. The network's average congestion is very low when 0% of the units are routed, while the central edges become extremely congested when 100% of the units are routed via the shortest route. Instead, the proposed game model and STACKELUCB algorithm allow the operator to select alternative routes depending on the users' demands, leading to improved congestion and a larger cumulative reward compared to the baselines.

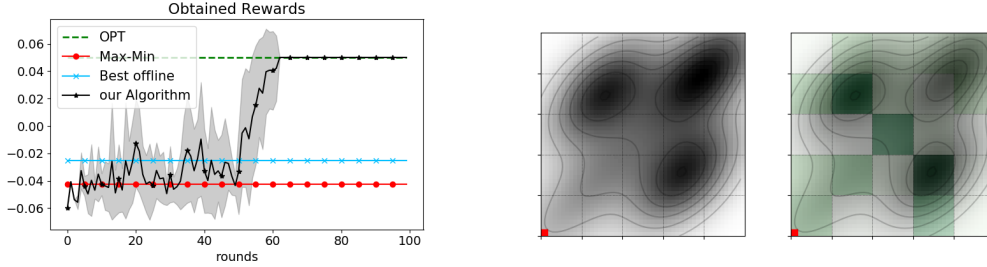

Figure 2: **Left:** Obtained rewards when the rangers know the poachers' model (OPT), assume the worst possible poaching location (Max-Min), estimate the poachers' model by using 1'000 offline data points (Best-offline), or use Algorithm (8) to update their patrol strategy online. Our algorithm discovers the optimal strategy in ~60 rounds and outperforms the considered baselines. **Right:** Park animal density (left plot) and rangers' mixed strategy (right plot, where probabilities are proportional to the green color intensity) computed with Algorithm (8). The poachers' model and starting location (red square) are not known by the rangers ahead of time.

## 4.2 Wildlife Protection against Poaching Activity

We consider a wildlife conservation task where the goal of park rangers is to protect animals from poaching activities. We model this problem as a sequential game between the rangers, who commit to a patrol strategy, and the poachers that observe the rangers' strategy to decide upon a poaching location [40, 18]. We study the repeated version of this game in which the rangers start with no information about the poachers' model and use Algorithm (8) to discover the best patrol strategy online.

We consider the game model of [18] that we briefly summarize below. The park area is divided into 25 disjoint cells (see Figure 2). A possible *patrol strategy* for the rangers is represented by the mixed strategy vector $x \in [0,1]^{25}$, where $x[i]$ represents the coverage probability of cell $i$. The poachers are aware of the rangers' patrol strategy and can use it to determine a poaching location $y \in \mathbb{R}^2$. Given patrol strategy $x$ and poaching location $y$, the expected utility of the rangers is $r(x,y) = \sum_{i=1}^{25} \left( x[i] \cdot R_i^r + (1 - x[i]) \cdot P_i^r \right) \cdot \mathbb{1}_i(y)$, where $\mathbb{1}_i(y) \in \{0,1\}$ indicates whether location $y$ belongs to cell $i$, $R_i^r > 0$ and $P_i^r < 0$ are reward and penalty for covering / not covering cell $i$, respectively. The poaching location is chosen based on the *Subjective Utility (SU)* model [26] $y = b(x) = \arg\max_y SU(\cdot, y)$, which we detail in Appendix F. The function $SU(x,y)$ trades-off the animal density at location $y$ (see right plots in Figure 2; here, such density was generated as a mixture of Gaussian distributions to simulate distinct high animal density areas), the distance between $y$ and the poachers' starting location (e.g., we use the starting location depicted as red square in Figure 2), and the rangers' coverage probabilities $x$. Based on this model, the goal of the rangers is to discover the optimal patrol strategy $x$ that maximizes $r(x, b(x))$, despite not knowing the poachers' response function $b(\cdot)$. This is an instance of the single type problem considered in Section 3.2.

We consider a repeated version of this game where, at each round, the rangers choose a patrol strategy $x_t$, obtain a noisy observation of the poaching location $y_t = b(x_t) + \epsilon_t$, and use this data to improve their strategy according to Algorithm (8). The decision set $\mathcal{X}$ of the rangers consists of 500 mixed strategies randomly sampled from the simplex and 25 pure strategies (i.e., covering a single cell with probability 1). We use the Màtern kernel defined over the vectors $(x, \bar{\theta})$ where $\bar{\theta} \in \mathbb{R}^{25}$ represents the maximal animal density in each of the park cells and can be interpreted as the single (and known) opponent's type. In Figure 2 (left plot), we compare the performance of our algorithm with the ones achieved by: 1) Optimal strategy (OPT) $x^\star = \arg\max_{x \in \mathcal{X}} r(x, b(x))$ with known poachers' model, 2) Max-Min, i.e, $x_\mathrm{m} = \arg\max_{x \in \mathcal{X}} \min_y r(x, y)$, which assumes the worst possible poaching location, and 3) Best-offline, that is, $x_\mathrm{o} = \arg\max_{x \in \mathcal{X}} r(x, \mu_\mathrm{o}(x))$, where $\mu_\mathrm{o}(\cdot)$ is the mean estimate of $b(\cdot)$ computed *offline* as in (2) by using 1'000 random data points. We average the obtained results over 10 different runs. Our algorithm outperforms the considered baselines and discovers the optimal patrol strategy after $\sim 60$ rounds. In Appendix F, we also show that our approach outperforms the standard GP bandit algorithm GP-UCB [35] which ignores the rewards' bi-level structure and directly tries to learn the function $g(\cdot) = r(\cdot, b(\cdot, \bar{\theta}))$. Finally, in Figure 2 (rightmost plot), we show the optimal strategy discovered by our algorithm despite not knowing the poachers' model (and starting location). We observe that the cells covered with higher probabilities are the ones with a high animal density near to the poachers' starting location.

# 5    Conclusions

We have considered the problem of learning to play repeated sequential games versus unknown opponents. We have proposed an online algorithm for the learner, when facing adversarial opponents, that attains sublinear regret guarantees by imposing kernel-based regularity assumptions on the opponents' response function. Furthermore, we have shown that our approach can be specialized to repeated Stackelberg games and demonstrated its applicability in experiments from traffic routing and wildlife conservation. An interesting direction for future work is to consider adding additional structure into opponents' responses by, e.g., incorporating bounded-rationality models of opponents as considered by [40] and [7].

## Broader Impact

Our approach is motivated by sequential decision-making problems that arise in several domains such as road traffic, markets, and security applications with potentially significant societal benefits. In such domains, it is important to predict how the system responds to any given decision and take this into account to achieve the desired performance. The methods proposed in this paper require to observe and quantify (via suitable indicators) the response of the system and to dispose of computational resources to process the observed data. Moreover, it is important that the integrity and the reliability of such data are verified, and that the used algorithms are complemented with suitable measures that ensure the safety of the system at any point in time.

### Acknowledgments

This work was gratefully supported by the Swiss National Science Foundation, under the grant SNSF 200021_172781, by the European Union's ERC grant $815943$, and ETH Zürich Postdoctoral Fellowship 19-2 FEL-47.

## Footnotes

[1] Our results also holds when $k(x, \theta, x', \theta') \leq L$ for some $L > 0$ (see Proof C for details).

[2]In accordance with this terminology, we use leader and follower to refer to learner and opponent, respectively.

[3]Unlike the previous section where $x_t$ belongs to a finite set $\mathcal{X}$, in this section, the set $\mathcal{X} = \Delta^{n_l}$ is infinite.

[4]In fact, Lipschitzness of $b(\cdot, \cdot)$ is implied by the RKHS norm bound assumption and certain properties of the used kernel function (see [10, Lemma 1] for details).

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
