[Supplementary Material]

# Supplementary Material

## Learning to Play Sequential Games versus Unknown Opponents

**Pier Giuseppe Sessa, Ilija Bogunovic, Maryam Kamgarpour, Andreas Krause (NeurIPS 2020)**

## A    RKSH Regression and Confidence Lemma

From the previously collected data $\{(x_i, \theta_i, y_i)\}_{i=1}^{t-1}$, a kernel ridge regression estimate of the opponent's response function can be obtained at every round $t$ by solving:

$$\underset{b \in \mathcal{H}_k}{\arg\min} \sum_{i=1}^{t-1} \left(b(x_i, \theta_i) - y_i\right)^2 + \lambda\|b\|_k \tag{10}$$

for some regularization parameter $\lambda > 0$. The representer theorem (see, e.g., [31]) allows to obtain a standard closed form solution to (10), which is given by:

$$\mu_t(x, \theta) = k_t(x, \theta)^T \left(K_t + \lambda I_t\right)^{-1} \boldsymbol{y}_t$$

where $\boldsymbol{y}_t = [y_1, \dots, y_t]^T$ is the vector of observations, $k_t(x, \theta) = [k(x, \theta, x_1, \theta_1), \dots, k(x, \theta, x_t, \theta_t)]^T$ and $[K_t]_{i,j} = k(x_i, \theta_i, x_j, \theta_j)$ is the kernel matrix. The estimate $\mu_t(\cdot, \cdot)$ can also be interpreted as the posterior mean function of the corresponding Bayesian Gaussian process model [31]. Similarly, one can also obtain a closed-form expression for the variance of such estimator, also interpreted as posterior covariance function, via the expression:

$$\sigma_t^2(x, \theta) = k(x, \theta, x, \theta) - k_t(x, \theta)^T \left(K_t + \lambda I_t\right)^{-1} k_t(x, \theta).$$

A standard result [2, 35], which forms the basis of ours and of many other Bayesian Optimization algorithms, shows that the functions $\mu_t(\cdot, \cdot)$ and $\sigma_t(\cdot, \cdot)$ can be used to construct confidence intervals that contain the true opponent's response function values with high probability. We report such result in the following main lemma, which states that given the previously observed opponent's actions, its response function belongs (with high probability) to the interval $[\mu_t(\cdot, \cdot) \pm \beta_t \sigma_t(\cdot, \cdot)]$, for a carefully chosen confidence parameter $\beta_t \geq 0$.

**Lemma 4** *Let $b \in \mathcal{H}_k$ such that $\|b\|_{\mathcal{H}_k} \leq B$ and consider the regularized least-squares estimate $\mu_t(\cdot, \cdot)$ with regularization constant $\lambda > 0$. Then for any $\delta \in (0, 1)$, with probability at least $1 - \delta$, the following holds simultaneously over all $x \in \mathcal{X}$, $\theta \in \Theta$ and $t \geq 1$:*

$$|\mu_t(x, \theta) - b(x, \theta)| \leq \beta_t \sigma_t(x, \theta),$$

*where $\beta_t = \sigma\lambda^{-1}\sqrt{2\log\left(\frac{1}{\delta}\right) + \log(\det(I_t + K_t/\lambda))} + \lambda^{-1/2}B$.*

### A.1    The case of multiple outputs

We consider the case of multi-dimensional responses $y_t = b(x_t, \theta_t) + \epsilon_t \in \mathbb{R}^m$, where $\{\epsilon_t[i], i = 1, \dots, m\}$ are i.i.d. and conditionally $\sigma$-sub-Gaussian with independence over time steps. In this case, posterior mean and variance functions can be obtained respectively as:

$$\mu_t(x, \theta) = \left[\mu_t(x, \theta, 1), \dots, \mu_t(x, \theta, m)\right]^T \quad , \quad \sigma_t^2(x, \theta) = \left[\sigma_t^2(x, \theta, 1), \dots, \sigma_t^2(x, \theta, m)\right]^T,$$

where $\mu_t(x, \theta, i)$ is the posterior mean estimate computed as in (2) using responses $\boldsymbol{y}_t = [y_1[i], \dots y_t[i]]^T$ and $\sigma_t^2(x, \theta, i)$ is the corresponding variance, for $i = 1, \dots, m$. Moreover, Lemma 4 shows that a careful choice of the confidence parameter $\beta_t$ implies that, with probability at least $1 - m\delta$, $|\mu_t(x, \theta, i) - b(x, \theta)[i]| \leq \beta_t \sigma_t(x, \theta, i)$ for any $x \in \mathcal{X}$, $\theta \in \Theta$, and $i = 1, \dots, m$. Hence, in this case the vector-valued functions $\mu_t(\cdot, \cdot)$ and $\sigma_t(\cdot, \cdot)$ can be used to construct a high-confidence upper and lower confidence bounds of the unknown function $b(\cdot, \cdot)$.

# B    Proof of Theorem 1

Our goal is to bound the learner's cumulative regret $R(T) = \max_{x \in \mathcal{X}} \sum_{t=1}^{T} r(x, b(x, \theta_t)) - \sum_{t=1}^{T} r(x_t, y_t)$, where $x_t$'s are the actions chosen by the learner and $y_t = b(x_t, \theta_t)$ is the opponent's response at every round $t$.

To bound $R(T)$, we first observe that the "optimistic" reward function $\tilde{r}_t(\cdot, \cdot)$ upper bounds the learner's rewards at every round $t$. Recall that for every $x \in \mathcal{X}$ and $\theta \in \Theta$, it is defined as:

$$\tilde{r}_t(x, \theta) := \max_y r(x, y)$$

$$\text{s.t.} \quad y \in \big[\text{lcb}_t(x, \theta), \text{ucb}_t(x, \theta)\big].$$

Moreover, according to Lemma 4, with probability $1 - \delta$ it holds:

$$\text{lcb}_t(x, \theta) \leq b(x, \theta) \leq \text{ucb}_t(x, \theta) \quad \forall x \in \mathcal{X}, \forall \theta \in \Theta, \quad \forall t \geq 1 \tag{11}$$

with $\text{lcb}_t(\cdot, \cdot)$ and $\text{ucb}_t(\cdot, \cdot)$ defined in (4) and setting $\beta_t$ as in Lemma 4. Therefore, conditioning on the event (11) holding true, by definition of $\tilde{r}_t(\cdot, \cdot)$ we have:

$$\tilde{r}_t(x, \theta) \geq r(x, b(x, \theta)) \quad \forall x \in \mathcal{X}, \forall \theta \in \Theta, \quad \forall t \geq 1. \tag{12}$$

By using (12) and defining $x^\star = \arg\max_{x \in \mathcal{X}} \sum_{t=1}^{T} r(x, b(x, \theta_t))$, the regret of the learner can now be bounded as:

$$R(T) = \sum_{t=1}^{T} r(x^\star, b(x^\star, \theta_t)) - \sum_{t=1}^{T} r(x_t, y_t)$$

$$\leq \sum_{t=1}^{T} \tilde{r}_t(x^\star, \theta_t) - \sum_{t=1}^{T} r(x_t, y_t)$$

$$= \underbrace{\sum_{t=1}^{T} \tilde{r}_t(x^\star, \theta_t) - \tilde{r}_t(x_t, \theta_t)}_{R_1(T)} + \underbrace{\sum_{t=1}^{T} \tilde{r}_t(x_t, \theta_t) - r(x_t, y_t)}_{R_2(T)},$$

where in the last equality we add and subtract the term $\sum_{t=1}^{T} \tilde{r}_t(x_t, \theta_t)$. We proceed by bounding the terms $R_1(T)$ and $R_2(T)$ separately.

We start by bounding $R_2(T)$. Let $y_t^\star = \arg\max_{y \in [\text{lcb}_t(x_t, \theta_t), \text{ucb}_t(x_t, \theta_t)]} r(x_t, y)$. Then, by definition of $\tilde{r}_t(\cdot, \cdot)$ we have

$$R_2(T) = \sum_{t=1}^{T} r(x_t, y_t^\star) - r(x_t, y_t) \leq L_r \sum_{t=1}^{T} \|(x_t - x_t, y_t^\star - y_t)\|_2$$

$$\leq L_r \sum_{t=1}^{T} |y_t^\star - y_t| \leq L_r \sum_{t=1}^{T} \big(\text{ucb}_t(x_t, \theta_t) - \text{lcb}_t(x_t, \theta_t)\big)$$

$$\leq 2 L_r \beta_T \sum_{t=1}^{T} \sigma_t(x_t, \theta_t) \leq 4 L_r \beta_T \sqrt{T \lambda \gamma_T}.$$

The first inequality follows from the Lipschitz continuity of $r(\cdot, \cdot)$, the second one is due to the event in (11) holding true, and the third one is by the definition of $\text{ucb}_t(\cdot, \cdot)$ and $\text{lcb}_t(\cdot, \cdot)$ and since $\beta_t$ is increasing in $t$. The last inequality follows since $\sum_{t=1}^{T} \sigma_t(\theta_t, x_t) \leq 2\sqrt{T \lambda \gamma_T}$ (see, e.g., Lemma 4 in [9]) for $\lambda \geq 1$ and assuming $k(\cdot, \cdot) \leq 1$. [5]

To complete the proof it remains to bound the regret term

$$R_1(T) = \sum_{t=1}^{T} \tilde{r}_t(x^\star, \theta_t) - \tilde{r}_t(x_t, \theta_t). \tag{13}$$

Note that $R_1(T)$ corresponds exactly to the regret that the learner incurs in an adversarial online learning problem in the case of sequence of reward functions $\tilde{r}_t(\cdot, \theta_t)$, $t = 1, \ldots, T$. Moreover, in Algorithm 1, the learner plays actions $x_t$'s according to the standard MW update algorithm which makes use of these functions in the form of full-information feedback.

Therefore, by using the standard online learning results (e.g., [8, Corollary 4.2]), if the learning parameter $\eta$ is selected as $\eta = \sqrt{\frac{8 \log |\mathcal{X}|}{T}}$ in the MW algorithm, then with probability at least $1 - \delta$,

$$R_1(T) \leq \sqrt{\tfrac{1}{2} T \log |\mathcal{X}|} + \sqrt{\tfrac{1}{2} T \log \left(\tfrac{1}{\delta}\right)}.$$

We remark that the above bound holds even when the rewards functions (in our case the types $\theta_t$'s) are chosen by an *adaptive* adversary that can observe the learner's randomized strategy $\mathbf{p}_t$ (see, e.g., [8, Remark 4.3]).

Having bounded $R_1(T)$, by using the standard probability arguments we obtain that with probability at least $(1 - 2\delta)$,

$$R(T) \leq \sqrt{\tfrac{1}{2} T \log |\mathcal{X}|} + \sqrt{\tfrac{1}{2} T \log \left(\tfrac{1}{\delta}\right)} + 4 L_r \beta_T \sqrt{T \lambda \gamma_T}.$$

## C  Proof of Corollary 2

For any sequence of types $\theta_t$'s and learner actions $x_t$'s, we follow the same proof steps as in proof of Theorem 1 to show that, with probability at least $1 - \delta$, the learner's regret can be bounded as

$$R(T) \leq \underbrace{\sum_{t=1}^{T} \tilde{r}_t(x^\star, \theta_t) - \tilde{r}_t(x_t, \theta_t)}_{R_1(T)} + \underbrace{\sum_{t=1}^{T} \tilde{r}_t(x_t, \theta_t) - r(x_t, y_t)}_{R_2(T)},$$

where $\tilde{r}_t(\cdot, \cdot)$ is the "optimistic" reward function defined in (6). Moreover, as we show in the proof of Theorem 2, $R_2(T) \leq 4 L_r \beta_T \sqrt{T \lambda \gamma_T}$ with probability at least $1 - \delta$.

Finally, we use the assumption $\theta_t = \bar{\theta}$, $\forall t \geq 1$, and the strategy in (8) to show that $R_1(T) \leq 0$. By assuming $\theta_t = \bar{\theta}$ for $t \geq 1$, we can write

$$R_1(T) = \sum_{t=1}^{T} \tilde{r}_t\left(x^\star, \bar{\theta}\right) - \tilde{r}_t\left(x_t, \bar{\theta}\right),$$

which is at most zero as the learner selects $x_t = \arg\max_{x \in \mathcal{X}} \tilde{r}_t\left(x, \bar{\theta}\right)$ at every round.

The corollary's statement then follows by observing that $R(T) \leq R_2(T)$ with probability at least $1 - \delta$.

## D  Proof of Corollary 3

As discussed in Section 3.3, in a repeated Stackelberg game the decision $x_t \in \Delta^{n_l}$ represents the leader's mixed strategy at round $t$, where $\Delta^{n_l}$ is the $n_l$- dimensional simplex. Hence, the regret of the leader can be written as

$$R(T) = \max_{x \in \Delta^{n_l}} \sum_{t=1}^{T} r(x, b(x, \theta_t)) - \sum_{t=1}^{T} r(x_t, y_t),$$

where $b(\cdot, \theta_t)$ is the best-response function of the follower of type $\theta_t$.

Before bounding the leader' regret, recall that the algorithm resulting from Corollary 3 consists of playing STACKELUCB over a finite set $\mathcal{D}$, which is a discretization of the leader's mixed strategy space $\Delta^{n_l}$. We choose $\mathcal{D}$ such that $\|x - [x]_{\mathcal{D}}\|_1 \leq \sqrt{n_l/T}/(L_r(1 + L_b))$ for every $x \in \Delta^{n_l}$, where $[x]_{\mathcal{D}}$ is the closest point to $x$ in $\mathcal{D}$. A natural way to obtain such a set $\mathcal{D}$ for the leader is to discretize the simplex $\Delta^{n_l}$ with a uniform grid of $|\mathcal{D}| = (L_r(1 + L_b)\sqrt{n_l T})^{n_l}$ points.

Define $x^\star = \arg\max_{x \in \Delta^{n_l}} \sum_{t=1}^T r(x, b(x, \theta_t))$, and let $[x^\star]_{\mathcal{D}}$ be the closest point to $x^\star$ in $\mathcal{D}$. Then, the leader's regret can be rewritten as:

$$R(T) = \sum_{t=1}^T r(x^\star, b(x^\star, \theta_t)) - \sum_{t=1}^T r(x_t, y_t)$$

$$= \underbrace{\sum_{t=1}^T r\big(x^\star, b(x^\star, \theta_t)\big) - r\big([x^\star]_{\mathcal{D}}, b([x^\star]_{\mathcal{D}}, \theta_t)\big)}_{R_A(T)} + \underbrace{\sum_{t=1}^T r\big([x^\star]_{\mathcal{D}}, b([x^\star]_{\mathcal{D}}, \theta_t)\big) - r(x_t, y_t)}_{R_B(T)}$$

where we have added and subtracted the term $\sum_{t=1}^T r\big([x^\star]_{\mathcal{D}}, b([x^\star]_{\mathcal{D}}, \theta_t)\big)$. At this point, note that the regret term $R_B(T)$ is precisely the regret the leader incurs with respect to the best point in the set $\mathcal{D}$. Therefore, since the points $x_t$ are selected by STACKELUCB over the same set, by Theorem 1 with probability at least $1 - 2\delta$,

$$R_B(T) \leq \sqrt{\tfrac{1}{2}T \log |\mathcal{D}|} + \sqrt{\tfrac{1}{2}T \log\left(\tfrac{1}{\delta}\right)} + 4L_r \beta_T \sqrt{T\lambda\gamma_T} \,. \tag{14}$$

The term $R_A(T)$ can be bounded using our Lipschitz assumptions on $r(\cdot)$ and $b(\cdot, \theta_t)$ as follows:

$$R_A(T) = \sum_{t=1}^T r(x^\star, b(x^\star, \theta_t)) - r([x^\star]_{\mathcal{D}}, b([x^\star]_{\mathcal{D}}, \theta_t))$$

$$= \sum_{t=1}^T r(x^\star, b(x^\star, \theta_t)) - r(x^\star, b([x^\star]_{\mathcal{D}}, \theta_t)) + r(x^\star, b([x^\star]_{\mathcal{D}}, \theta_t)) - r([x^\star]_{\mathcal{D}}, b([x^\star]_{\mathcal{D}}, \theta_t))$$

$$\leq \sum_{t=1}^T L_r \|x^\star - [x^\star]_{\mathcal{D}}\|_1 + L_r \|b(x^\star, \theta_t) - b([x^\star]_{\mathcal{D}}, \theta_t)\|_1$$

$$\leq \sum_{t=1}^T L_r \|x^\star - [x^\star]_{\mathcal{D}}\|_1 + L_r L_b \|x^\star - [x^\star]_{\mathcal{D}}\|_1$$

$$= \sum_{t=1}^T L_r(1 + L_b)\|x^\star - [x^\star]_{\mathcal{D}}\|_1 \leq \sum_{t=1}^T L_r(1 + L_b)\frac{\sqrt{n_l/T}}{L_r(1 + L_b)} = \sqrt{n_l T} \,.$$

In the first inequality we have used $L_r$-Lipschitzness of $r(\cdot)$, in the second one $L_b$-Lipschitzness of $b(\cdot, \theta_t)$, and the last inequality follows by the property of the constructed set $\mathcal{D}$.

The statement of the corollary then follows by summing the bounds of $R_A(T)$ and $R_B(T)$ and substituting in (14) the cardinality $|\mathcal{D}| = (L_r(1 + L_b)\sqrt{n_l T})^{n_l}$.

# E   Experimental setup of Section 4.1

In this section, we describe the experimental setup of Section 4.1. First, we explain how we generated the set of routing plans $\mathcal{X}$ for the network operator, and the demand profiles $\theta_t$'s for the other users in the network. Then, we detail how the network congestion level $y_t$ is determined as a function of the operator's plan and the users' demand profiles. Finally, we summarize the rest of the parameters chosen for our experiment.

We generate a finite set $\mathcal{X}$ of possible routing plans for the operator as follows. The operator can decide to route $0\%, 25\%, 50\%, 75\%$, or $100\%$ of the 300 units from origin to destination (blue and green nodes in Figure 1); moreover, the routed units can be split in 3 groups of equal size, and each group can take a potentially different route among the 3 shortest routes from origin to destination. This results in a total of $|\mathcal{X}| = 41$ possible plans for the operator. At each round $t$, the plan chosen by the operator is represented by the occupancy vector $x_t \in \mathbb{R}^{|E|}_{\geq 0}$ indicating how many units are routed through each edge of the network (see Section 4.1).

We use the demand data from [21, 37] to build the users' demand profile $\theta_t \in \mathbb{R}^{552}_{\geq 0}$ at each round, indicating how many users want to travel between any two nodes of the network (it represents the *type* of opponent the operator is facing at round $t$). This data consists of units of demands associated

with $24 \cdot 23 = 552$ origin-destination pairs. Each entry $\theta_t[i]$ is obtained by scaling the demand corresponding to the origin-destination pair $i$ by a random variable uniformly distributed in $[0,1]$, for $i = 1, \ldots, 552$.

Given operator's plan $x_t$ and demands $\theta_t$, in Section 4.1 we modeled the averaged congestion over the network edges with the relation

$$y_t = b(x_t, \theta_t).$$

The function $b(\cdot, \cdot)$ includes 1) the network congestion model and 2) how the users choose their routes in response to the operator's plan $x_t$. Below, we explain in detail these two components.

**Congestion model.** Congestion model and related data are taken from [21, 37]. Data consist of nodes' 2-D positions and edges' capacities and free-flow times, while the congestion model corresponds to the widely used used Bureau of Public Roads (BPR) model. The congestion in the network is determined as a function of the edges' occupancy (i.e., how many units traverse each edge), which can be represented by the *occupancy vector* $z \in \mathbb{R}_{\geq 0}^{|E|}$. Then, according to the BPR model, the travel time to traverse a given edge $e \in E$ increases with the edge's occupancy $z[e] \in \mathbb{R}_{\geq 0}$ following to the relation:

$$t_e(z) = c_e \cdot \left[1 + 0.15\left(\frac{z[e]}{C_e}\right)^4\right] \quad e = 1, \ldots |E|, \tag{15}$$

where $c_e$ and $C_e$ are free-flow time and capacity of edge $e$, respectively.

In our example, given routing plan $x_t$ of the network operator and routes chosen by the other users (below we explain how such routes are chosen as a function of $x_t$), we can compute the occupancy vector at round $t$ as

$$z_t = x_t + u_t,$$

where the vector $u_t \in \mathbb{R}_{\geq 0}^{|E|}$ represents the network occupancy due to the users ($u_t[e]$ indicates how many users are traveling trough edge $e$, $e = 1 \ldots |E|$). Hence, according to the BPR model, we define

$$c_e(z_t) = 0.15\left(\frac{z_t[e]}{C_e}\right)^4 \quad e = 1, \ldots |E|, \tag{16}$$

to be the *congestion* of edge $e$ at round $t$. It represents the extra (normalized) time needed to traverse edge $e$. Using (16), the averaged congestion over the network edges $y_t \in \mathbb{R}_+$ is computed as

$$y_t = \frac{1}{|E|} \sum_{e \in E} c_e(z_t). \tag{17}$$

**Users' preferences.** Given routing plan $x_t$ chosen by the network operator, the users choose routes as follows. We consider the two shortest routes (in terms of distance) between any two nodes in the network. Then, we let the users select the route with minimum travel time among the two, where the travel time of each edge is $t_e(x_t)$, computed as in (15). That is, users choose the routes with minimum travel time, assuming the occupancy of the network is the one caused by the operator.

In our experiment, the operator obtains a noisy observation of $y_t$, where the noise standard deviation is set to $\sigma = 5$. Moreover, we set the trade-off parameter $\kappa = 10$ for the operator's objective, in order to obtain meaningful trade-offs. Finally, in our experiments we scale by a factor of $0.01$ both the demands and the edges' capacities taken from [21, 37].

# F  Supplementary material for Section 4.2

We provide additional details and experimental results for the wildlife conservation task considered in Section 4.2.

## F.1  Poachers' model and response function

Here, we more formally describe the Subjective Utility model [26] for the poachers and hence the poachers' response function used in the experiment.

When poaching at location $y$, the poachers obtain reward [18]:

$$R^p(y) = \phi(y) - \zeta \cdot \frac{D(y)}{\max_y D(y)}, \tag{18}$$

Figure 3: Obtained rewards when the rangers know the poachers' model (OPT), use the proposed algorithm to update their patrol strategy online (**Left**), or use GP-UCB ignoring the bi-level rewards' structure (**Right**), for different choices of the confidence parameter $\beta_t$. When the confidence $\beta_t$ is sufficiently small, the proposed algorithm consistently discovers the optimal strategy in $\sim$60 rounds, while GP-UCB either converges to suboptimal solutions or experiences a slower learning curve.

where $\phi : \mathbb{R}^2 \to [0, 1]$ is the park animal density function (see right plots in Figure 2 where $\phi(\cdot)$ was generated as a mixture of Gaussian distributions), $D(y)$ is the distance between $y$ and the poachers' starting location (we use the starting location depicted as red square in Figure 2), and $\zeta$ is a trade-off parameter measuring the importance that poachers give to $D(y)$ compared to $\phi(y)$. Using (18), the expected utility of the poachers (unknown to the rangers) follows the Subjective Utility (SU) model [26]:

$$SU(x, y) = \sum_{i=1}^{25} \Big( - \omega_1 f(x[i]) + \omega_2 R^p(y) + \omega_3 P_i^p \Big) \cdot \mathbb{1}_i(y) \,,$$

where $f$ is the S-shaped function $f(p) = (\delta p^\gamma)/(\delta p^\gamma + (1 - p)^\gamma)$ from [18], $R^p(y)$ is the reward for poaching at location $y$, $P_i^p < 0$ is a penalty for poaching in cell $i$, and the coefficients $\omega_1, \omega_2, \omega_3 \geq 0$ describe the poachers' preferences. Given a patrol strategy $x$, hence, we assume that the poachers select location $y = b(x) = \arg\max_y SU(x, y)$ to maximize their own utility function. [6]

For the poachers' utility we use $w_1 = -3, w_2 = w_3 = 1, \delta = 2, \gamma = 3, \zeta = 0.5, P_i^p = -1$, while we set $R_i^r = 1, P_i^r = -\phi(y)$ for the rangers' reward function.

### F.2 Additional experimental results

We provide additional experimental results comparing the performance of the proposed algorithm, which learns the response function $b(\cdot, \hat{\theta})$ and exploits the bi-level structure of the reward function, with the one of GP-UCB [35] (standard baseline for GP bandit optimization) which learns directly the function $g(\cdot) = r(\cdot, b(\cdot, \theta))$. We run both algorithms using a Màtern kernel, with kernel hyperparameters computed offline data via a maximum likelihood method over 100 random data points. To run our algorithm we set noise standard deviation $\sigma$ to 2% of the width of the park area, while for GP-UCB we set $\sigma$ to 2% of the rewards' range. In Figure 3 we compare the performance of the two algorithms for different choices of the confidence parameter $\beta_t$. For sufficiently small values of $\beta_t$, the proposed approach consistently converges to the optimal solution in $\sim$60 iterations, while GP-UCB either converges to suboptimal solutions or displays a slower learning curve.

## Footnotes

[5]In case we have $k(\cdot, \cdot) \leq L$ for some $L > 0$ then the result holds for $\lambda \geq L$.

[6]In the case of more than one best response, ties are broken in an arbitrary but consistent manner.