[Reviews · NeurIPS 2020]

Review 1

Summary and Contributions: The paper studies a repeated sequential game where, at each iteration, a learner plays first and, then, an opponent responds to the leaners’ action. The paper considers the case in which the learner plays according to a mixed (randomized) strategy, while the opponent responds to the realized action according to a response function that is unknown to the learner. At each iteration, the learner faces an opponent of a different type, determining its response function, and, after playing, the learner observes the opponent’s type and its response (subject to some noise). The goal of the leaner is to maximize a known reward function of the chosen action and the opponents response. Thus, in the repeated setting, the learner performance is evaluated by means of the regret computed with respect to the best action in hindsight, knowing the realized sequence of opponent’s types and their actual response functions. The paper introduces a no-regret algorithm (called StackelUCB) based on a multiplicative-weight update formula computed with respect to some optimistic estimates of the learner’s reward function (using UCB-style bounds on the opponent’s response functions values). Then, a regret bound for the algorithm is provided, when the response functions satisfy some regularity/smoothness assumptions. The paper also considers the setting of Stackelberg repeated games (a special case in which the learner’s action space is a simplex) and conducts an experimental evaluation of the StackelUCB algorithm on two real-world-inspired scenario: a traffic routing game and a wildlife conservation task.

Strengths: I believe that the model and results presented in the paper are interesting, especially for an audience from the algorithmic game theory and online learning communities. While the repeated Stackelberg game setting has been already studied in the literature, the paper generalizes current state of the art along two directions: (i) it considers a more general model that could encompass other bi-level online optimization problems; and (ii) it addresses the case in which the opponent’s response functions are not known in advance, which makes the setting much more interesting from a practical perspective. As for the experimental evaluation, the paper does a good job in comparing the performance of the proposed algorithm with different baselines on structured game instances representing interesting real-world applications.

Weaknesses: While the modeling assumptions adopted in the paper are reasonable and standard when dealing with repeated games in an online learning scenario, I think that the paper does not spend enough words in justifying the fact that the learner can observe the type of the opponent at the end of each iteration. This is a reasonable assumption from a theoretical perspective, even though, differently from what is stated in the paper (Lines 171-172 for instance), it makes the feedback received by the learner something between a full-information setting (in which the learner must also know the opponent’s response functions) and a partial-information setting (in which the learner cannot observe the opponent’s type). I would have appreciated a more detailed discussion on this point (see also questions below). Furthermore, the derivation of the regret bounds, while correct, is obtained by using straightforward arguments and piggybacking on other known results.

Correctness: As far as I am concerned all the results and proofs are correct.

Clarity: Overall, the paper is well written and organized. The following are some suggestions on how to improve the readability and correct some typos (in order of appearance): Line 25: “…, and thus, …” -> “…, and, thus, …”. Lines 56-57: It is not clear at this point that the sequence of opponent’s types is not only chosen adversarially in advance, but it may also be chosen by an adaptive adversary that observes the sequence of strategies employed by the leaner in the past and the corresponding sequence of realized actions (up to iteration t-1). Line 69: “In every round…” -> “In every round $t$…”. Line 95: The notation after “such that…” is not clear. Line 185: Use “$i$” instead of “$\tau$” as suffix like in the rest of the paper.

Relation to Prior Work: The paper does a good job in comparing with other works.

Reproducibility: Yes

Additional Feedback: I have the following questions to the authors: - Can you provide some intuitions on how to deal with a setting with completely-partial (in the sense that the learner does not even observe the opponent’s type) feedback? - Can you provide more intuition as to why your regret bounds (see e.g. Theorem 1) do not depend on the size of the opponent’s type space? This is not the case in state-of-the-art results on repeated Stackelberg games, where the dependence on the number of types appears in the regret bound (see e.g. [3] in your references).


Review 2

Summary and Contributions: The paper considers a problem where two players, a learner and an opponent, are playing a sequential game, with the opponent's type changing in every round and opponent's response function for any type unknown to the learner. The opponent is shortsighted once its type is determined, i.e., only respond to the learner's current round action, which will be observed by the opponent before he takes action. However, the opponent's type can be adversarially chosen and may depend on the learner's previous actions, which represents the worst case the learner may face. The paper has a few contributions. (1) The authors proposed a UCB-based algorithm to prescribe the learner's strategy in each round. (2) They show the algorithm leads to no regret. (3) They specialize their proposed approach and regret bound to Stackelberg games. (4) They tested the algorithms in simulation settings inspired by two problem domains, traffic routing and wildlife conservation, and showed clearly superior performance compared to baselines.

Strengths: The paper is relevant to NeurIPS community and the problem studied is interesting and important. I think the model they focus on is nicely formulated and it is indeed captures a problem in many real-world challenges. The use of real-world problem inspired simulation settings is appropriate. The proposed algorithm is well grounded and leads to significantly better performance in the experiments.

Weaknesses: Some aspects of the model need to be better clarified or justified. For example, the opponent's type may depend on not only the learner's previous actions but also the learner's overall strategy (the learning algorithm per se), right? If so, in some sense, there are two levels of leader-follower type of interaction. In each round, the learner first takes an action and the opponent responds. In the overall game, the learner chooses a policy first and then the opponent responds by choosing its type in each round to harm the learner. The current description in Sec 2 does not highlight this aspect. The assumption that the learner observes the opponent's type is a very strong one. It is unclear why this makes sense in real world problems.

Correctness: There is no obvious errors. The experiments are reasonable.

Clarity: The paper is nicely written in general. It would be better if the RKHS part can be stated more clearly. In line 95, the current diescription of $b(x,\theta)$ is confusing. It may be helpful to provide some details in the appendix and refer the readers to the appendix. It may be good to highlight early in the paper (say, Sec 3.1) that EXP3 can be directly applied to this problem, but it ignores the observed opponent type and thus is not fully exploiting the problem structure. Also, it would be good to explain how restrictive the regularity assumptions and the assumptions on noise term are. Typo: Line 109, an extra comma in the definition of $\mathcal{H}_t$. Line 153, $r(.)$->$r(\cdot)$

Relation to Prior Work: The paper provides a reasonable coverage of related work and discussion of the differences.

Reproducibility: Yes

Additional Feedback: After rebuttal: I thank the authors for providing clarifications and explanations.


Review 3

Summary and Contributions: A new algorithm for no-regret learning in "sequential games" (Stackelberg games) with varying opponent types is proposed and the no-regret property is developed formally under limiting assumptions. The algorithm is evaluated empirically in simulated traffic routing and wildlife conservation tasks, showing improvements over several baselines. UPDATE AFTER REBUTTAL I thank the authors for clarifying some of my comments. I'm adding a few additional comments below in case they are useful. Re randomising opponents: Are you sure there is no advantage in randomising *for the opponent*? I wonder whether there might be cases in which the opponent could randomise to confuse the learner in some situations with a useful outcome for the opponent, or to hide the opponent's intentions. If you are referring to a specific formal result, it could help to remind the reviewer of that. Re observing opponent type: While I see the authors' point that observing types can be useful, it still strikes me that not many applications will easily provide this information. Most reviewers have noted this as a strong assumption that requires justification in the paper.

Strengths: The formal development of this work is rigorous, which is nice. I also appreciate the empirical evaluation on top of the formal aspects, which shows improved performance over some baseline algorithms.

Weaknesses: The paper makes a number of limiting assumptions. For example, while the learner agent is allowed to randomise, the opponent is not. Why? Moreover, it is assumed that the learner gets to see the opponent's type after each interaction. How is this assumption justified? There are also some "regularity assumptions", such as that the opponent is a member of a kernel Hilbert space induced by some a-priori known positive-definite kernel function, and that the learner's reward function is Lipschitz continuous in L1. How limiting/realistic are these assumptions in practice? In Section 3 an additional assumption is made that opponent responses must be real-valued (not discrete-valued). In my view these assumptions limit the significance and applicability of this work. The practicality of these assumptions should be discussed in paper. The paper uses "regret" as the guiding principle. However, regret is known to have important conceptual limitations. For example, minimising regret does not in general equate to maximising reward. See e.g. Crandall, Jacob W. "Towards minimizing disappointment in repeated games." Journal of Artificial Intelligence Research 49 (2014): 111-142. I believe good reasons should to be given to justify the use of regret as an optimisation objective. The evaluation in traffic routing is interesting. However, much work exists in this domain and I think it would be important to compare with some existing solutions in this space, as opposed to comparing to only some other bandit algorithms like Hedge and Exp3. I'm unable to assess significance of results in Fig1 table since no variance metric is reported (std dev or std err of mean) and no statistical hypothesis testing was done. In the wildlife task, why is the proposed algorithm no longer compared to other bandit algorithms as was done in the traffic task? Here, none of the baselines are learning algorithms.

Correctness: Formal derivations seem correct.

Clarity: Assumptions made in this work are sprinkled across the different sections. For example, the noisy scalar assumption is only revealed in Sec 3. For clarify such assumptions should be listed in one place.

Relation to Prior Work: In introduction and rest of paper, the authors refer to their class of games as "sequential games" which I find is a misnomer. The considered games are (a type of) Stackelberg games in which one player (leader, learner) chooses first, after which a second player (follower, opponent) chooses next. Indeed the authors call the proposed algorithm "StackelUCB". In contrast, the term sequential games suggests to me a much broader class of games such as extensive-form games, stochastic games, Dec-POMDPs, POSGs etc; these include a very large body of work on opponent modelling not mentioned here. For example, the authors claim that prior work focussed on settings in which opponents have one fixed type, which is not correct. There is a whole class of algorithms called "type-based reasoning" which can handle different opponent types. For example, see the HBA algorithm [1] and I-POMDPs [2]. For a broader positioning in opponent modelling, I recommend the Albrecht/Stone survey [3]. [1] S. Albrecht, J. Crandall, S. Ramamoorthy. Belief and Truth in Hypothesised Behaviours. Artificial Intelligence (AIJ), Vol. 235, pp. 63-94, 2016 [2] Gmytrasiewicz, P., Doshi, P., 2005. A framework for sequential planning in multiagent settings. Journal of Artificial Intelligence Research 24 (1), 49–79. [3] S. Albrecht, P. Stone. Autonomous Agents Modelling Other Agents: A Comprehensive Survey and Open Problems. Artificial Intelligence (AIJ), Vol. 258, pp. 66-95, 2018 The introduction also cites several works on opponent modelling in multi-agent reinforcement learning. In the MARL setting the agents' policies change throughout time (non-stationarity) and hence these methods

Reproducibility: Yes

Additional Feedback: As another possible evaluation domain, I'd like to suggest the airport security domains used in the works of Tambe et al.


Review 4

Summary and Contributions: This paper studies a repeated game played between a learner and its opponent. The learner does not know the type of the opponent that she is playing with, nor the full information about their best-response function. The paper takes a regret-minimization approach and presents an algorithm that provides sublinear regret guarantees. They tested their approach with a routing problem and a Stackelberg security game and showed that their approach outperforms existing online learning algorithms and other benchmarks in these tasks.

Strengths: The paper is technically sound, with detailed analysis and results with theoretical guarantees. The contribution, as far as I'm concerned, lies in applying techniques from Bayesian optimization to solving this online learning problem, whereby their approach to modelling the opponent captures a wide range of best-response behaviors without relying on more specific assumptions about the opponent's utilities. This is novel and the paper is relevant to the NeurIPS community.

Weaknesses: 1. Some related work seems to have been overlooked: Playing Repeated Security Games with No Prior Knowledge (Xu et al, AAMA 2016). As far as I can see, the paper also makes no assumption about the opponent's utilities, so it seems to be an important benchmark to compare with. (In fact, they didn't even make the assumption that the opponent is behaving according to some best-response function, so perhaps more legitimate to say that in their model the opponent model is unknown.) Following the above comment, I wonder how effective the proposed approach is in regret minimization, compared with an approach similar to that of Xu et al. Understandably, the regret is defined slightly differently here as the best-response function is also incorporated in the regret, so the goal is not completely the same, but in any case, it seems necessary to have some comparison, at least empirically, to highlight the advantage of the proposed approach. 2. I'm not sure how efficient and practical the approach is in dealing with repeated Stackelberg games. The discretization of the leader's mixed strategy space seems to be very costly, so the scalability of the approach is questionable.

Correctness: I don't find any errors. However, I have limited knowledge about some of the techniques used in the paper.

Clarity: The paper is well-written. Some technical parts may not be very accessible to readers out of the specific research area, so some discussion about the underlying idea and intuition behind the proposed approach might be helpful.

Relation to Prior Work: The paper has clearly discussed the related literature and compared their work with previous papers; though it seems to have missed some related work (see comments for "Weaknesses").

Reproducibility: Yes

Additional Feedback: In the experiments, different kernels were used. How does the choice of kernels affect the performance of the approach?

[Author Response · NeurIPS 2020]

We would like to thank all the reviewers for their constructive feedback. In the following, we respond (**R**) to individual concerns (**C**) summarized in italic. Citations refer to references in the paper.

**Reviewer 1. C:** *"Can you provide some intuitions on how to deal with a setting with completely-partial (in the sense that the learner does not even observe the opponent's type) feedback?"* **R:** In such a setting (also denoted as 'bandit feedback') the learner could play according to the Exp3 algorithm [2], as discussed in Lines 137-140. Compared to StackelUCB, the reward estimates obtained by Exp3 do not exploit of the rewards bilevel structure, yielding a higher variance and an unavoidable $\mathcal{O}(\sqrt{|\mathcal{X}|})$ term in the resulting regret bound, as discussed in Line 169. **C:** *"- Can you provide more intuition as to why your regret bounds (see e.g. Theorem 1) do not depend on the size of the opponent's type space?"* **R:** Compared to state-of-the-art results (e.g., [3]), our regret bounds do not depend on the number of types, but only on the *dimension* of the corresponding type space $\Theta$, via the maximum information gain $\gamma_T$. For instance, in case of the squared exponential kernel we have $\gamma_T = \mathcal{O}((\log T)^{d+1})$, where $d$ is the dimension of $\mathcal{X} \times \Theta$. This is because, compared to [3], our algorithm can exploit the present correlations among different types (i.e., the fact that similar types lead to similar opponent responses) through the RKHS model.

**Reviewer 2. C:** *"...the opponent's type may depend on not only the learner's previous actions but also the learner's over-all strategy (the learning algorithm per se), right?"* **R:** Indeed, the sequence of types can be chosen by an adaptive adversary who knows the learner's past actions and the learner's algorithm (but not the realization of its internal randomization). We will make sure to better emphasize that our model accommodates this fact. **C:** *"The assumption that the learner observes the opponent's type is a very strong one. It is unclear why this makes sense in real world problems."* **R:** We agree that observing the opponent's type is a stronger assumption than the standard bandit feedback, however in some applications (such as the ones studied in our experimental section) one may receive information about the opponent a-posteriori, that can be utilized to improve the playing strategy. In the considered traffic example, for instance, the network operator can reconstruct the past demands in the network. In security domains, one may acquire information about the attacker after an attack has taken place (e.g., as in [3]). This information can be encoded as opponent's *type*, and our work shows that it can significantly improve the learner's performance (when available) compared to only using the bandit feedback.

**Reviewer 3. C:** *"...while the learner agent is allowed to randomise, the opponent is not. Why? "* **R:** We considered deterministic responses since the opponent plays second, i.e., only after observing the learner's play, and hence there are no advantages in considering randomized strategies for the opponent. **C:** *"The practicality of these assumptions should be discussed in paper"* **R:** The main assumptions of our model are observing the opponent's types and assuming its response function has a small RKHS norm. Observing opponent's types is of practical interest, e.g., in security domains (see response to Reviewer 2), and a key contribution of our work is to show that such observations can significantly improve the learner performance. Assuming a small RKHS norm is a typical non-parametric assumption used in black-box Bayesian optimization to efficiently learn and optimize an unknown function by lifting it to a higher dimensional feature space. It has found several practical relevance during the past years (see, e.g., [30, 34]). The optimal kernel choice is problem-specific, although squared exponential kernels have universal approximation properties. **C:** *"In the wildlife task, why is the proposed algorithm no longer compared to other bandit algorithms as was done in the traffic task? Here, none of the baselines are learning algorithms."* **R:** In the wildlife task the learner faces a single type of opponent and hence this leads to different algorithmic benchmarks than the traffic routing task. The Reviewer is correct in that Figure 2 compares our method only against offline strategies, however we have also considered the GP-UCB [34] bandit algorithm as a natural learning benchmark, as discussed in Section 4.2. A direct comparison with GP-UCB, under different learning rates, is included in Appendix F due to space limitations.
Finally, we thank the Reviewer for pointing out the relevant literature on "opponent modeling" and "type-based reasoning" which we will include in the paper. We have identified our setup as a general 'sequential game' since the key component is learner and opponent playing sequential moves, the second observing the action of the first but not necessarily playing a best-response function (as in Stackelberg games). We will clarify the distinctions and connections with the mentioned literature result in the paper.

**Reviewer 4. C:** *"Some related work seems to have been overlooked: Playing Repeated Security Games with No Prior Knowledge (Xu et al, AAMA 2016)."* **R:** We thank the Reviewer for bringing up this related work and we will add a reference and discussion in our paper. We would like to point out that such work focuses on playing repeated *security* games, i.e., where the learner's reward structure and corresponding feedback information follow the specific combinatorial model of allocating security resources to protect a given set of targets. The Follow-the-Perturbed Leader (FPL) online learning algorithm proposed by Xu et al. exploits this specific combinatorial structure. When applied to our general sequential games framework, such FPL-based algorithm essentially corresponds to the bandit Exp3 [2] (see Lines 137-140) which we have compared both theoretically (see discussion after Theorem 1) and experimentally (see Section 4.1) with the proposed method. **C:** *'In the experiments, different kernels were used. How does the choice of kernels affect the performance of the approach?"* **R:** In general, we observed that certain kernels are more suitable than others depending on the application. In the traffic experiment we observed similar performance with polynomial kernels of different degrees and with squared exponential kernels (which have the property of being universal approximators), while in the wildlife example we experienced similar results with Matérn kernels with different hyperparameters.

[Meta-Review · NeurIPS 2020]

Four knowledgeable referees all recommended that the paper be accepted. The reviewers appreciated the many answers provided by the author’s in the rebuttal. After the rebuttal, the reviewers did continue to have concerns about (among other things) the strong assumptions made in the paper. While I agree with the reviewers in these concerns, they and I both concur that the paper makes a publishable contribution in NeurIPS. I recommend that the paper be accepted for poster presentation. I recommend that the authors address the clarifications they promised in their reviews along with the other points made by the reviewers.